# Prognostic Value of Troponin Elevation in COVID-19 Hospitalized Patients

**DOI:** 10.3390/jcm9124078

**Published:** 2020-12-17

**Authors:** Elena-Mihaela Cordeanu, Nicolas Duthil, Francois Severac, Hélène Lambach, Jonathan Tousch, Lucas Jambert, Corina Mirea, Alexandre Delatte, Waël Younes, Anne-Sophie Frantz, Hamid Merdji, Valérie Schini-Kerth, Pascal Bilbault, Patrick Ohlmann, Emmanuel Andres, Dominique Stephan

**Affiliations:** 1Department of Hypertension, Vascular Disease and Clinical Pharmacology, Strasbourg Regional University Hospital, 67091 Strasbourg, France; nicolas.duthil@chru-strasbourg.fr (N.D.); helene.lambach@chru-strasbourg.fr (H.L.); jonathan.tousch@chru-strasbourg.fr (J.T.); corina.mirea@chru-strasbourg.fr (C.M.); anne-sophie.frantz@chru-strasbourg.fr (A.-S.F.); dominique.stephan@chru-strasbourg.fr (D.S.); 2Division of Public Health, Methodology and Biostatistics, University Hospitals of Strasbourg, 67091 Strasbourg, France; francois.severac@chru-strasbourg.fr; 3Department of Vascular Medicine, Mulhouse Regional Hospital, 68100 Mulhouse, France; jambertlucas@gmail.com; 4Department of Cardiology, Haguenau Regional Hospital, 67500 Haguenau, France; delatte.alex@gmail.com; 5Department of Vascular Medicine, Colmar Regional Hospital, 68000 Colmar, France; wael.younes@ch-colmar.fr; 6Intensive Care and Reanimation Department, Strasbourg Regional University Hospital, 67091 Strasbourg, France; hamid.merdji@chru-strasbourg.fr; 7UMR 1260 INSERM Regenerative Nanomedecine, Faculty of Pharmacy, Strasbourg University, 67400 Illkirch, France; valerie.schini-kerth@unistra.fr; 8Emergency Department, Strasbourg Regional University Hospital, 67091 Strasbourg, France; pascal.bilbault@chru-strasbourg.fr; 9Cardiology Department, Strasbourg Regional University Hospital, 67091 Strasbourg, France; patrick.ohlmann@chru-strasbourg.fr; 10Internal Medicine Department, Strasbourg Regional University Hospital, 67091 Strasbourg, France; emmanuel.andres@chru-strasbourg.fr

**Keywords:** COVID-19, troponin, myocardial injury, SARS-CoV-2, cardiovascular, biomarker

## Abstract

(1) Background: Severe acute respiratory syndrome coronavirus 2 (SARS-CoV-2) penetrates the respiratory epithelium through angiotensin-converting enzyme-2 (ACE2) binding. Myocardial and endothelial expression of ACE2 could account for the growing body of reported evidence of myocardial injury in severe forms of Human Coronavirus Disease 2019 (COVID-19). We aimed to provide insight into the impact of troponin (hsTnI) elevation on SARS-CoV-2 outcomes in patients hospitalized for COVID-19. (2) Methods: This was a retrospective analysis of hospitalized adult patients with the SARS-CoV-2 infection admitted to a university hospital in France. The observation period ended at hospital discharge. (3) Results: During the study period, 772 adult, symptomatic COVID-19 patients were hospitalized for more than 24 h in our institution, of whom 375 had a hsTnI measurement and were included in this analysis. The median age was 66 (55–74) years, and there were 67% of men. Overall, 205 (55%) patients were placed under mechanical ventilation and 90 (24%) died. A rise in hsTnI was noted in 34% of the cohort, whereas only three patients had acute coronary syndrome (ACS) and one case of myocarditis. Death occurred more frequently in patients with hsTnI elevation (HR 3.95, 95% CI 2.69–5.71). In the multivariate regression model, a rise in hsTnI was independently associated with mortality (OR 3.12, 95% CI 1.49–6.65) as well as age ≥ 65 years old (OR 3.17, 95% CI 1.45–7.18) and CRP ≥ 100 mg/L (OR 3.62, 95% CI 1.12–13.98). After performing a sensitivity analysis for the missing values of hsTnI, troponin elevation remained independently and significantly associated with death (OR 3.84, 95% CI 1.78–8.28). (4) Conclusion: Our study showed a four-fold increased risk of death in the case of a rise in hsTnI, underlining the prognostic value of troponin assessment in the COVID-19 context.

## 1. Introduction

Human Coronavirus Disease 2019 (COVID-19), resulting from a newly described respiratory viral infection with severe acute respiratory syndrome coronavirus 2 (SARS-CoV-2), was originally identified in December 2019 in Wuhan, China, before becoming a global pandemic. Cardiovascular risk factors and underlying cardiovascular diseases were associated with worse prognosis [1]. Moreover, cardiovascular involvement in COVID-19 has been related to the viral infective mechanism through the binding of the spike envelope protein to cell membrane angiotensin-converting enzyme-2 (ACE2) [2,3]. ACE2 is physiologically implicated in balancing the deleterious effects of renin–angiotensin system activation, being highly expressed by the lungs, kidneys, gut, and brain and has also been identified in the cardiovascular system [4]. Endothelial and myocardial expression of ACE2 could account for myocardial injury, defined by a rise in troponin (Tn) associated with some severe forms of COVID-19 [5,6]. In March 2020, the American College of Cardiology suggested Tn measurement in the context of COVID-19 only if myocardial infarction was suspected [7]. Since then, myocardial injury has been described in approximately one-third of COVID-19 patients [8,9,10,11,12,13]. Recent publications have associated an elevation in cardiac and inflammatory biomarkers with infection severity and worse prognosis [13,14]. Whether cardiac biomarkers such as Tn could have a prognostic value in SARS-CoV-2 is still under debate, and Tn measurement is not systematically performed in infected patients. We report herein a retrospective analysis of hospitalized adult patients, in whom a high-sensitivity troponin I (hsTnI) test was performed, from a university hospital in Eastern France, one of the most affected areas in Europe during the first wave of the COVID-19 pandemic.

## 2. Experimental Section

### 2.1. Study Design and Patient Selection

We performed a retrospective analysis of electronic medical records of hospitalized COVID-19 patients admitted to the University Hospital of Strasbourg between 25 February 2020 (date of admission of the first case) and 1 April 2020. The study was approved by the Strasbourg University Hospital Ethical Committee. All patients aged more than 18 years old were selected on the basis of laboratory-confirmed COVID-19 infection by positive reverse-transcriptase polymerase chain reaction (RT-PCR) on a nasopharyngeal swab. A local RT-PCR kit was used to detect SARS-CoV-2. The observation period ended at discharge. Vital status at discharge was known for all hospitalized patients. Thus, all patients having had at least one measurement of hsTnI during hospitalization were included in the first analysis and the entire cohort in the sensitivity analysis.

### 2.2. Baseline Variables

Data concerning medical history, chronic medication, clinical presentation, laboratory findings, and low-dose pulmonary computed tomography (CT) lesions were collected. hsTnI test results were retrieved from two hospital sites with site- and sex-specific 99th percentile upper reference limits (URLs) (Hospital Site 1: Siemens Advia Centaur XP and XPT assay with URLs of 37 ng/L for women and 57 ng/L for men; Hospital Site 2: Siemens Dimension Vista assay with URLs of 54 ng/L for women and 79 ng/L for men). When several tests were performed for the same patient, maximum levels of hsTnI were recorded. Given the infectivity risk and according to guidelines, echocardiography was only sporadically performed, and data were colligated. Antiviral, antibiotic, and anticoagulant treatments during hospitalization were equally reported.

### 2.3. Outcome Assessment

For the purpose of this study, the observation period ended at hospital discharge, with a median length of stay of 16.5 days (interquartile range (IQR): 8–29). All patient data were collected during hospitalization. The main evaluation criterion was in-hospital death from any cause. Recourse to high-flow nasal oxygen (HFNO) therapy, noninvasive ventilation (NIV), orotracheal intubation (OTI), and occurrence of severe sepsis, arterial or venous thrombosis, and acute renal impairment were colligated. Events were abstracted from clinical charts or discharge synopsis. The evaluation criteria were adjudicated by senior physicians of the vascular medicine unit.

### 2.4. Statistical Analysis

This was a retrospective cohort study; therefore, no power calculation was performed. Continuous variables were expressed as mean ± standard deviation (SD) or median with interquartile range (IQR), depending on their distribution. The normality of the distribution was assessed using the Shapiro–Wilk test. Categorical variables were presented as numbers of cases (percentages/frequencies). Continuous variables were compared using the Student’s *t*-test or Wilcoxon rank-sum test, for categorical variables, Fisher’s exact tests were employed. In order to address potential sources of bias, clinically pertinent risk factors associated with mortality in univariate analysis were selected as candidates for the multivariate logistic regression analysis. Results were expressed as odds ratios (ORs) with 95% confidence intervals (CI). A sensitivity analysis including patients without troponin measurement was performed using multiple imputation by chained equation to handle missing values [15,16]. A value of *p* < 0.05 was considered statistically significant. All analyses were performed using R software version 3.2.2 (R Foundation for Statistical Computing, Vienna, Austria. URL https://www.R-project.org/).

## 3. Results

### 3.1. Patients Characteristics at Baseline

A total of 943 COVID-19 patients were admitted to the University Hospital of Strasbourg from 25 February 2020 to 1 April 2020, of whom 375 (67% of males, mean age of 66 ± 14.4 ranging from 21 to 93 years) were included in this analysis after exclusion of patients hospitalized for less than 24 h (*n* = 145), minors (*n* = 14), patients hospitalized for other medical reasons and incidentally found positive for SARS-CoV-2 PCR (*n* = 12), and patients without a hsTnI test (*n* = 397) (Figure 1).

The remaining population was divided into two subgroups based on hsTnI elevation, namely “elevated hsTnI” (*n* = 126) and “normal hsTnI” (*n* = 249). A comprehensive, comparative description of baseline characteristics and in-hospital outcomes for patients having an elevated hsTnI versus normal hsTnI test is presented in Table 1. A rise in troponin was associated with higher age and comorbid conditions such as pre-existing hypertension, diabetes, dyslipidemia, chronic heart failure, and chronic kidney disease. In-hospital treatment (antiviral, antibiotics, and anticoagulation) did not differ between groups (Table 1).

### 3.2. Troponin Elevation and in Hospital Outcomes

A rise in troponin was associated with unfavorable outcomes such as severe sepsis, acute renal impairment, stroke, and death (Table 1, Figure 2).

### 3.3. Mortality Predictors

In the multivariate logistic regression analysis between survivors and nonsurvivors, including variables statistically significant in univariate analysis and deemed clinically pertinent, mortality was independently associated with an age of at least 65 years old (OR 3.17, 95% CI 1.45–7.18), C-reactive protein (CRP) elevation of at least 100 mg/L (OR 3.62, 95% CI 1.12–13.98), and hsTnI elevation (OR 3.12, 95% CI 1.49–6.65) (Table 2).

After performing a sensitivity analysis on the entire cohort including patients without hsTnI measurement, age of at least 65 years old (OR 4.99, 95% CI 2.69–9.25), active cancer (OR 2.52, 95% CI 1.22–5.22), chronic kidney disease (OR 3.26, 95% CI 1.85–5.76), CRP elevation of at least 100 mg/L (OR 2.34, 95% CI 1.28–4.28), D-dimer elevation over 3000 µg/L (OR 1.95, 95% CI 1.04–3.52), and hsTnI elevation (OR 3.84, 95% CI 1.78–8.28) were significantly associated with death (Appendix A).

## 4. Discussion

### 4.1. Troponin Elevation and in-Hospital Mortality

In our study, a rise in hsTnI was identified in 34% of the cohort. Patients with hsTnI elevation had a higher prevalence of “cardiovascular burden,” translating into a four-fold increased risk of death compared to normal hsTnI patients (47% vs. 12%). Furthermore, when performing a sensitivity analysis for missing values of hsTnI, a rise in troponin remained independently associated with death. Our findings are consistent with recently published data showing that cardiac biomarkers such as Tn, B-type natriuretic peptide, or D-dimer are commonly elevated in hospitalized COVID-19 patients [14]. However, a rise in Tn appears more accurate in predicting mortality compared to other biomarkers [14,17]. In a meta-analysis of 16 studies, Zou et al. found a pooled overall incidence of myocardial injury of 24.4% (542/2224) and a markedly increased all-cause mortality associated with a rise in Tn (72.6% vs. 14.5%) [12]. Furthermore, in the largest study published to date on Tn as a predictor of death including 6247 COVID-19 patients, Majure et al. observed significantly increased death rates in the case of Tn elevation compared to normal Tn levels (43% vs. 13%) [13].

Troponin assays differ among studies and details concerning cut-offs and manufacturers were not systematically reported in the literature [18]. Nonetheless, Tn remains an objective marker easily obtainable with an obvious prognosis impact. Indeed, Manocha et al. showed in a cohort of 446 patients that high Tn levels were a potent predictor of 30-day in-hospital mortality with an adjusted OR of 4.38, allowing the authors to develop and validate a mortality score including age, hypoxia, and Tn elevation beyond the 75th percentile (HA2T2) to predict death [14].

TheACS-related rise in hsTnI in our study was modest (2.3%) and is consistent with other retrospective studies reporting similar observations. Thus, Tn appears to have low specificity for ACS in the COVID-19 context, but myocardial injury could serve as a severity marker rather than being a cause of death per se [13,14,19].

### 4.2. Myocardial Injury in COVID-19: Definition and Potential Mechanisms

Although most studies defined myocardial injury on the sole basis of Tn elevation, Giustino et al. performed an extensive study on echocardiographic abnormalities and found cardiac structural alterations in two-thirds of patients with enzymatic rise, which was strongly correlated to mortality [20]. The recent perception of COVID-19 as a systemic condition acting through multiple deleterious mechanisms, such as inflammation, endothelial dysfunction, prothrombotic state, and myocardial injury, could explain the severe course of the disease in some patients. The hypothetical mechanisms of myocardial injury include tissular hypoxia, cytokine storm, and direct viral myocardial lesions [1,12,18].

### 4.3. Limitations

Given the retrospective nature of our study, only patients with a hsTnI measurement were included in the first analysis, which is susceptible to selection bias. However, performing a sensitivity analysis for missing values allowed us to confirm the initial results, adding robustness to our observation. Moreover, the use of two troponin measurement assays with different URLs did not allow a ROC-curve cut-off calculation. Nonetheless, high-sensitivity troponin tests are recognized as being more accurate than previous measurement techniques and are the recommended assay for assessing myocardial injury defined by an increase in hsTnI over the 99th percentile URL. As hsTnI URLs differ between assays, compiling data using hsTnI as a continuous variable could be biased. For this reason, we consider that using hsTnI elevation as a categorical variable was a practical alternative, palliating sex- and site-specific URLs and responding to guidelines’ myocardial injury definition.

### 4.4. Further Implications

Although current ICU approaches have marginally changed between the first and second waves of the epidemics, potentially affecting the generalizability of our conclusions, there is still no available prognosis-changing antiviral treatment. Thus, given the considerable size of our population, we deem that our results could be pertinent for presently hospitalized patients as Tn is one of the earliest markers of end-organ dysfunction reflecting relative hypoxia [13,21]. As such, we advocate for Tn assessment to improve risk stratification and prompt more intensive surveillance with potential therapeutic consequences on enhancing tissular perfusion.

## 5. Conclusions

Elevated troponin measurement in COVID-19 hospitalized patients was independently correlated to in-hospital mortality and could serve as a predictor for short-term survival providing an indication for intensive clinical vigilance and potentially improving patients’ triage.

## Figures and Tables

**Figure 1 jcm-09-04078-f001:**
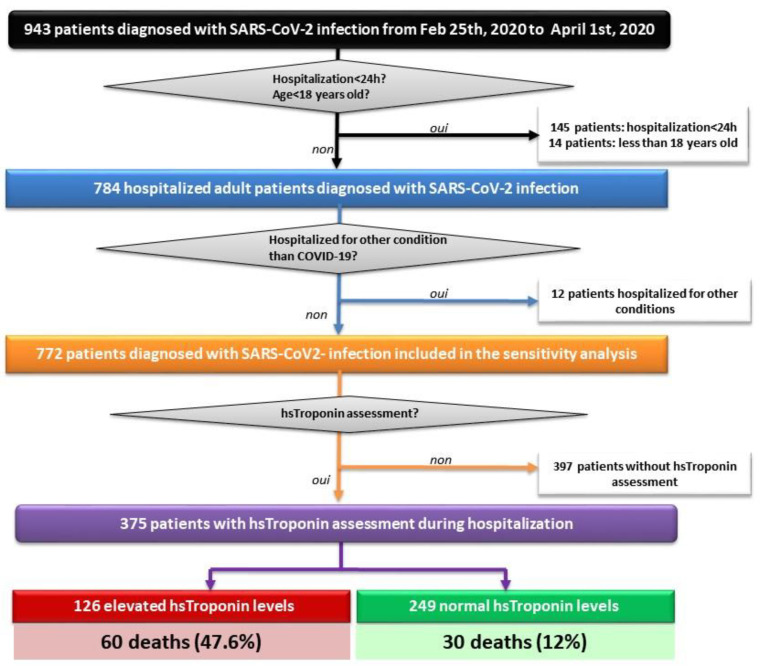
Study flowchart showing patient selection. COVID-19: Human Coronavirus Disease 2019; Feb: February; h: hours; hs: high-sensitivity; SARS-CoV-2: severe acute respiratory syndrome coronavirus 2.

**Figure 2 jcm-09-04078-f002:**
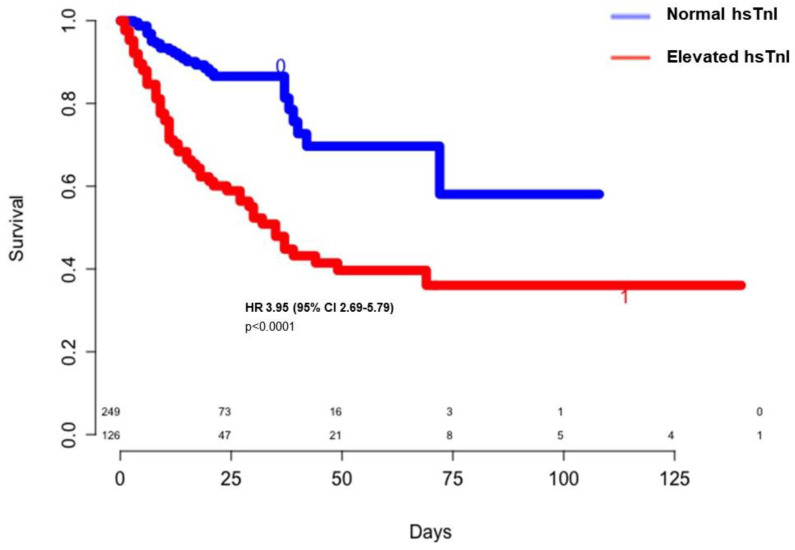
Crude survival rates according to high-sensitivity troponin levels. CI: confidence interval; HR: hazard ratio; hsTnI: high-sensitivity troponin I.

**Table 1 jcm-09-04078-t001:** Baseline characteristics and in-hospital outcomes.

	Overall Cohort N (%)/M(IQR)	Elevated hsTnI N (%)/M(IQR)	Normal hsTnI N (%)/M(IQR)	*p*-Value	SMD
***n***	**375**	**126**	**249**		
**Age (years)**	66 (55.5–74)	71.5 (64.2–79)	63 (51–71)	<0.001	0.627
Age ≥ 65 years old	210 (56)	94 (74.6)	116 (46.6)	<0.001	0.598
**Male**	252 (67.2)	82 (65.1)	170 (68.3)	0.61	0.076
**BMI (kg/m^2^)** (*n* = 337)	28 (25–32)	29 (26–34)	28 (25–31)	0.04	0.035
**eGFR (mL/min/1.73 m^2^) on admission**	81 (50–96)	55.5 (28–84)	86 (70–100)	<0.001	0.816
**Cardiovascular risk factors**					
Hypertension (*n* = 374)	221 (58.9)	96 (76.2)	125 (50.2)	<0.001	0.575
Diabetes (*n* = 374)	126 (33.6)	59 (46.8)	67 (26.9)	<0.001	0.430
Dyslipidemia (*n* = 374)	144 (38.4)	66 (52.4)	78 (31.3)	<0.001	0.446
Smoking (history or current) (*n* = 330)	80 (21.3)	30 (23.8)	50 (20.1)	0.36	0.121
Obesity (*n* = 347)	134 (35.7)	51 (40.5)	83 (33.3)	0.20	0.122
**Medical history**					
Heart disease (*n* = 374)	65 (17.3)	31 (24.6)	34 (13.7)	0.007	0.305
Ischemic heart disease	48 (12.8)	21 (16.7)	27 (10.8)	0.15	0.173
Chronic heart failure	20 (5.3)	12 (9.5)	8 (3.2)	0.009	0.263
HFrEF	13 (3.5)	7 (5.5)	6 (2.4)	0.20	0.478
Chronic kidney disease (*n* = 374)	65 (17.3)	32 (25.4)	33 (13.2)	0.003	0.316
Chronic respiratory disease (*n* = 374)	45 (12)	16 (12.7)	29 (11.6)	0.73	0.305
Active cancer	18 (4.8)	9 (7.1)	9 (3.6)	0.20	0.157
Cognitive impairment (*n* = 374)	15 (4)	7 (5.5)	8 (3.2)	0.19	0.117
VTE (*n* = 374)	22 (5.9)	11 (8.7)	11 (4.4)	0.07	0.177
**Admission treatment**					
**Antithrombotic treatment**	121 (32.3)	55 (43.7)	66 (26.5)	<0.001	0.387
Antiplatelet	89 (23.7)	37 (29.4)	52 (20.9)	0.007	0.212
Anticoagulation	41 (10.9)	23 (18.3)	18 (7.2)	<0.001	0.351
**Antihypertensive drugs**					
RASi	150 (40)	63 (50)	87 (34.9)	0.006	0.308
Diuretics	83 (22.1)	41 (32.5)	42 (16.9)	<0.001	0.387
Beta-blockers	103 (27.5)	47 (37.3)	56 (22.5)	<0.001	0.354
**Admission Low-dose chest CT ***	320 (85.3)	102 (81)	218 (87.6)	1	0.009
abnormal	314 (98.1)	100 (98)	245 (98.4)	1	
**COVID-19 infection severity indicators**					
Oxygen therapy flow rate of >5 L/min	231 (67.9)	82 (77.4)	149 (63.7)	0.01	0.304
ICU admission	215 (57.6)	85 (67.5)	130 (52.6)	0.008	0.307
Intubation	205 (54.8)	84 (66.7)	121 (48.8)	0.001	0.353
HFNO therapy/NIV	10 (2.7)	1 (0.8)	9 (3.6)	0.17	
CT scan extension > 25% (*n* = 320)	179 (47.7)	69 (54.8)	110 (44.2)	0.052	0.655
CRP ≥ 100 mg/L (*n* = 371)	270 (72)	106 (84.1)	164 (65.9)	<0.001	0.458
D-dimer count ≥ 3000 µg/L (*n* = 292)	170 (45.3)	77 (61.1)	93 (37.3)	<0.001	0.546
Lymphopenia < 1000/µL (*n* = 371)	284 (75.7)	104 (82.5)	180 (72.2)	0.026	0.276
**In-hospital treatment**					
Prophylactic/therapeutic anticoagulation	329 (88.2)	108 (86.4)	221 (89.1)	0.55	
Antibiotics	331 (88.3)	109 (86.5)	222 (89.2)	0.49	
Antiviral ^$^	199 (53.1)	60 (47.6)	139 (55.8)	1	
**In-hospital outcomes**					
Death	90 (24)	60 (47.6)	30 (12)	<0.001	0.844
Severe sepsis or septic shock	98 (27.2)	47 (37.9)	51 (21.6)	<0.001	0.389
Acute renal impairment	136 (36.3)	77 (61.1)	59 (23.7)	<0.001	0.851
VTE	57 (15.2)	25 (19.8)	32 (12.9)	0.10	0.190
Stroke/TIA	16 (4.3)	10 (7.9)	6 (2.4)	0.026	0.289
**Hospital length of stay (days)**	15 (8–29)	16.5 (8–35)	15 (8–27)	0.26	0.277

BMI: body mass index; CRP: C-reactive protein; CT: computer tomography; eGFR: estimated glomerular filtration rate; HFrEF: heart failure with reduced ejection fraction; hs-cTnI: high-sensitivity cardiac troponin I; HFNO: high-flow nasal oxygen; ICU: intensive care unit; IQR: interquartile range; M: median; N: number; NHF: nasal high flow; NIV: noninvasive ventilation; PCR: polymerase chain reaction; RASi: renin–angiotensin system inhibitor; SMD: standardized mean difference; VTE: venous thromboembolism; TIA: transient ischemic attack. * number (frequencies) of low-dose CT scans performed in each group. ^$^ remdesivir, lopinavir/ritonavir, oseltamivir, interferon, hydroxychloroquine

**Table 2 jcm-09-04078-t002:** Univariate and multivariate analysis of baseline risk factors for death.

Risk Factor	Unadjusted OR (95% CI)	*p*-Value	Adjusted OR (95% CI)	*p*-Value
Age ≥ 65 years old	5.55 (3.12–10.47)	<0.001	3.17 (1.45–7.18)	0.004
High blood pressure	2.08 (1.25–3.55)	0.005	0.78 (0.28–2.10)	0.63
Diabetes mellitus	1.66 (1.01–2.71)	0.04	0.95 (0.40–2.15)	0.90
Dyslipidemia	1.60 (0.98–2.59)	0.054	0.54 (0.22–1.25)	0.16
Tobacco consumption	2.33 (1.33–4.04)	0.002	1.99 (0.89–4.49)	0.09
Active Cancer	5.52 (2.10–15.47)	<0.001	2.80 (0.67–11.65)	0.15
Chronic kidney disease	3.07 (1.74–5.41)	<0.001	2.30 (0.84–6.41)	0.10
Ischemic heart disease	2.39 (1.25–4.50)	0.007	1.02 (0.30–3.36)	0.96
Chronic heart failure	5.39 (2.15–14.22)	<0.001	0.93 (0.17–4.86)	0.94
Previous antithrombotic drug	2.52 (1.54–4.13)	<0.001	1.72 (0.70–4.26)	0.23
Previous RASi	1.61 (1.00–2.60)	0.049	1.40 (0.55–3.61)	0.47
Lymphopenia	2.35 (1.25–4.77)	0.011	3.05 (0.90–14.35)	0.10
CRP ≥ 100 mg/L (max)	3.26 (1.71–6.76)	<0.001	3.62 (1.12–13.98)	0.042
D-dimer count (max) ≥ 3000 µg/L	2.52 (1.42–4.62)	0.002	1.55 (0.70–3.55)	0.28
hsTroponin elevation	6.63 (3.98–11.25)	<0.001	3.12 (1.49–6.65)	0.003

CI: confidence interval; CRP: C-reactive protein; hs: high-sensitivity; max: maximum; OR: odds ratio; RASi: renin–angiotensin system inhibitor.

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
