# Peer review of "Prognostic Value of Troponin Elevation in COVID-19 Hospitalized Patients"

_jcm, 2020, doi:10.3390/jcm9124078_

Round 1
Reviewer 1 Report
The paper presents a severe bis as the authors did not describe the assay(s) used to measure cardiac troponin and looking the differences in the cut off proposed and used in the two different centers it's clearly evident that data cannot be put together
Author Response
Strasbourg, the 30th of November 2020
Dear Reviewers,
Dear Editors,
We are pleased to submit to the Journal of Clinical Medicine a revised version of the manuscript “Prognostic value of troponin elevation in COVID-19 hospitalized patients” (Manuscript IDjcm-1015266) by Cordeanu and coworkers.
Our answers to reviewers’ comments appear in dark blue whereas all changes in the manuscript appear in red.
Reviewer(s)' Comments to Author :
- Reviewer 1.
Reviewer 1 comment 1 : The paper presents a severe bis as the authors did not describe the assay(s) used to measure cardiac troponin and looking the differences in the cut off proposed and used in the two different centers it's clearly evident that data cannot be put together.
Authors’ answer : We agree with reviewer’s 1 remarks as knowing the troponin assay that has been used is a quality guarantee for its interpretation. Moreover, high-sensitivity troponin assays have superior diagnosis performance when compared with previous generation troponin assays. High sensitivity troponin I upper reference limits (URLs) vary according to the manufacturer. As such, we have modified in the chapter “Experimental section” the following statement (changes appear in red):
“hsTnI test results were retrieved from two hospital sites with site- and sex-specific 99th percentile upper reference limits (URLs) (Hospital site 1: Siemens Advia Centaur XP and XPT assay with URLs of 37ng/l for women and 57ng/l for men; hospital site 2: Siemens Dimension Vista assay with URLs of 54ng/l for women and 79ng/l for men).
Concerning the possibility of analyzing data obtained with two assays from the same manufacturer but with different URLs, we believe that taking into account
the specific URLs and considering the troponin as a 2-dimension variable : normal or elevated, allows us to compile data, not as a continuous variable but a categorical one. Indeed, putting together data and constructing a ROCcurve could have presented a bias related to low but positive hsTnI levels according to one assay and negative according to the other. For instance a hsTnI of 60ng/L in a man is considered elevated according to assay 1 and normal according to assay 2. The same obstacle would also interfere with compiling men and women’s data as all hsTnI assays have sex-specific URLs. As such we estimate that having site- and sex- specific URLs is not a handicap as we took into consideration the defined URLs representing the 99e percentile in order to define elevated hsTnI. Moreover, having sex specific URLs for high sensitivity troponin is considered quality marker as it has been stated by several authors.
To our acceptance, compiling data of hsTnI performed by two different assays by transforming a quantitative variable into qualitative one (elevated/normal hsTnI) and respecting site and sex specific URLs does not constitue a bias of interpretation.
Thus, the 99th percentile URL is considered as the cut-off for detection of myocardial injury and diagnosis of myocardial infarction. International guidelines recommended that the increase in cTnI or cTnT levels over the 99th percentile upper reference limit (99th URL) should be considered as clinically relevant (The 2018 Fourth Universal Definition of MI - ESC/ACC/AHA/WHF Expert Consensus Document; IFCC Task Force Recommendations ). For this reason, we consider that using hsTnI elevation as a categorical variable is a good option, palliating sex and site specific URLs and responding to guidelines’ myocardial injury definition.
Furthermore, internal and external quality evaluation of our two assays showed equivalence on Youden’s diagram and an excellent correlation with r2 of 0.90 (see below).
Reviewer 2 comment 1: Introduction: How does your paper differ from the others that you cite that evaluate the association between troponin and survival?
Authors’answer: The association between mortality and elevated troponin in the covid context has already been described.. However, our study endorses several important features:
- Large study population
- Follow-up until discharge for all patients
- Showing that troponin is an Independent marker of mortality in both initial analysis and sensitivity analysis
- specific data concerning French population
Even if the relationship between troponin and mortality in COVID-19 has already been described, accumulating proofs are mandatory in reinforcing previous publications. We believe that our study is not redundant with previous published works on the matter, as novelty not only resides in the singularity of the observation, but also its robustness given by the amount of cumulating evidence.
Reviewer 2 comment 2 & 3: Methods : Until what follow-up date did you have updated vital status? This should be reported. Why did you only include patients with known discharge status?
Authors’answer: Vital status was updated until hospital discharge for all patients which was indirectly proven by the study flowchart which indicated that no patient was excluded on the basis of unknown discharge status. However, we agree with the reviewer that the formulation could be unclear.
As such, the following statement was reformulated:
“Thus, all patients for whom discharge status was known, i.e., either death during hospitalization or survival to discharge…were included.”
becoming:
“The observation period ended at discharge. Vital status at discharge was known for all hospitalized patients. Thus, all patients having had at least one measurement of hsTnI during hospitalization, were included in the main analysis and the entire cohort in the sensitivity analysis.”
Un update concerning the sensitivity analysis appears in the
Reviewer 2 comment 4: Why did you use a logistic regression analysis and not a time-to-event analysis and censor patients at last known follow-up?
Authors’answer: As all patients’vital status was known at discharge and the follow-up period was short (corresponding to the hospitalization length of stay), we deemed that a logistic regression was informative and a time to event analysis was not necessary as all events were censored.
Reviewer 2 comment 5: . How does the exclusion of patients with missing values, potentially bias your results? Did you consider performing sensitivity analyses with imputed values?
Authors’answer: The retrospective nature of our study exposes us to an important bias related to missing values for troponin. Indeed, 51 % of the hospitalized patients did not have a hsTnI evaluation. As such, we have integrated reviewer’s suggestion and performed a sensitivity analysis that appears now in the Supplementary Table 1. This sensitivity analysis reinforces the initial observation of troponin as an independent variable associated to mortality.
Reviewer 2 comment 6: Why did you choose to dichotomize troponin levels as opposed to evaluating the association between troponin level as a continuous variable and the outcome?
Authors’answer: We chose to use this dichotomization for two reasons :
- The use of two different assays with sex- and site-specific URLs (see Answer to reviewer 1 Question)
- Myocardial injury is defined by guidelines as an elevation of hsTn above URL cut-offs (see Answer to reviewer 1 Question)
This choice was also explained in the subsection “Limits” of the Discussion chapter as follows:
“Given the retrospective nature of our study, only patients with a hsTnI measurement were included in the first analysis which is susceptible of selection bias. However, performing a sensitivity analysis for missing values allowed us to confirm the initial results adding robustness to our observation. Moreover, the use of two troponin measurement assays with different URLs did not allow a ROC-curve cut-off calculation. Nonetheless, high-sensitivity troponin tests are recognized as being more accurate than previous measurement techniques and are the recommended assay for assessing myocardial injury defined by an increase in hsTnI over the 99th percentile URL. As hsTnI URLs differ between assays, compiling data using hsTnI as a continuous variable could be biased. For this reason, we consider that using hsTnI elevation as a categorical variable was a practical alternative, palliating sex- and site- specific URLs and responding to guidelines’ myocardial injury definition.”
Reviewer 2 comment 7: P values in Table 1 are potentially misleading. Significant values will occur by chance alone in one out of every 20 hypothesis tests. With large enough sample sizes you may see a "statistically significant" difference without seeing a clinically significant difference (e.g. BMI wtih a p=0.04 but not a marked difference in BMI).
Authors’answer: Indeed, when multiple comparisons are performed, p-values may be misleading. As such we have added the calculation of the standardized mean difference to Table 1. Furthermore, variables such as BMI for which the difference between groups although statically significant was not clinically pertinent were not included in the multivariate regression model.
****
Other changes were operated in order to include the new information brought by the sensitivity analysis (main text, abstract, key words, tables). Moreover, an error in the management of missing values for the variables included in the multivariate analysis was dected and corrections were made in the text and Table 2 accordingly.
Several other changes were operated in order to improve wording quality and English spelling.
In the revised manuscript, all changes appear in red :
page 1 – lines 4, 13-14, 34-35, 40-42, 43, 47-49
page 2 – lines 50, 81-84, 89-90
page 3 – line 100, 112-113
page 4 – lines 132-133
pages 4-5 - Table 1, column “SMD”, line 136
page 5 – lines 142-144, 151-155
pages 5-6 – table 2 columns “Adjusted OR” and “p-value”
page 6 – 159-163, 166, 168-169, 173, 175, 179
page 7 – lines 187, 192, 196, 199-208, 221, 223-225
page 8 – lines 229-230
page 9- lines 270-273, 274, 277, 280, 282, 284
One author was added as his contribution to data analysis (F.S.) needed to be recognized.
Two references (15,16) were added and the order of two others changed (18-19).
The corresponding author is: Elena-Mihaela Cordeanu
With our best regards.
Sincerely yours.
Elena-Mihaela CORDEANU, MD Prof Dominique STEPHAN, MD, PhD

Reviewer 2 Report
- Introduction
- How does your paper differ from the others that you cite that evaluate the association between troponin and survival?
- Methods
- Until what follow-up date did you have updated vital status? This should be reported.
- Why did you only include patients with known discharge status?
- Why did you use a logistic regression analysis and not a time-to-event analysis and censor patients at last known follow-up?
- Results
- How does the exclusion of patients with missing values, potentially bias your results? Did you consider performing sensitivity analyses with imputed values?
- Why did you choose to dichotomize troponin levels as opposed to evaluating the association between troponin level as a continuous variable and the outcome?
- P values in Table 1 are potentially misleading. Significant values will occur by chance alone in one out of every 20 hypothesis tests. With large enough sample sizes you may see a "statistically significant" difference without seeing a clinically significant difference (e.g. BMI wtih a p=0.04 but not a marked difference in BMI).
Author Response

(The authors gave the same response as above.)

Round 2
Reviewer 1 Report
I read with interest the new version of the paper and the answers of the authors to previous criticisms.
Despite the limitations due to study design (retrospetive and observational) the data presented confirm the value of hsTroponin I in the prognostication of COVID19 severity and risk of death.
While this is not a new insights, the confirmation value of the study makes the paper suitable for a publication as the authors made great efforts in answering to previously raised criticisms
Author Response
Strasbourg, the 14th of December 2020
Dear Reviewers,
Dear Editors,
We are pleased to submit to the Journal of Clinical Medicine a revised version of the manuscript “Prognostic value of troponin elevation in COVID-19 hospitalized patients” (Manuscript ID jcm-1015266) by Cordeanu and coworkers.
Reviewer(s)' Comments to Author :
- Reviewer 1 comment 1: I read with interest the new version of the paper and the answers of the authors to previous criticisms. Despite the limitations due to study design (retrospetive and observational) the data presented confirm the value of hsTroponin I in the prognostication of COVID19 severity and risk of death. While this is not a new insights, the confirmation value of the study makes the paper suitable for a publication as the authors made great efforts in answering to previously raised criticisms
- Reviewer 2 comment 1: The authors have adequately addressed my concerns.
Authors’ answer : We thank Reviewers for the positive comments and we are grateful for previous criticisms that allowed us to improve the analysis and bring robustness to our conclusions in despite of the obvious limits of the study related to its observational retrospective nature.
In addition, we have performed a correction in the references’ numbers.
Page 3 – line 114 : references 15 and 16 instead of 1 and 2
The corresponding author is: Elena-Mihaela Cordeanu
With our best regards.
Sincerely yours.
Elena-Mihaela CORDEANU, MD Prof Dominique STEPHAN, MD, PhD

Reviewer 2 Report
The authors have adequately addressed my concerns.
Author Response

(The authors gave the same response as above.)
